# Written Briefing and Oral Counseling Increase the Willingness to Receive the SARS-CoV-2 Vaccination among Women in Puerperium: A Qualitative Prospective Cohort Study

**DOI:** 10.3390/vaccines10091505

**Published:** 2022-09-09

**Authors:** Nawa Schirwani, Petra Pateisky, Tamina Koren, Alex Farr, Herbert Kiss, Dagmar Bancher-Todesca

**Affiliations:** Department of Obstetrics and Gynecology, Division of Obstetrics and Feto-Maternal Medicine, Medical University of Vienna, 1090 Vienna, Austria

**Keywords:** SARS-CoV-2, vaccination, pregnancy, puerperium

## Abstract

(1) Background: Vaccination rates for severe acute respiratory syndrome-coronavirus-2 (SARS-CoV-2) are low in Austria. International obstetric societies recommend the SARS-CoV-2 mRNA vaccination for women in puerperium. (2) Methods: A prospective two-stage cohort study was conducted at the Medical University of Vienna between October 2022 and December 2022. Firstly, women in puerperium were assigned to the evaluation group (step 1), and secondly, another cohort of unvaccinated women were randomly assigned to study group A (written briefing) or B (written and oral briefing) (step 2). We evaluated the vaccination status among women in the evaluation group and the willingness to receive the vaccination in all three cohorts. (3) Results: We included 217 women in puerperium (evaluation: n = 69, A: n = 68; B: n = 80). In the evaluation group, 66.7% (n = 46/69) of the women were unvaccinated. A total of 45.7% (21/46) of the unvaccinated women categorically declined the SARS-CoV-2 vaccination. A total of 26.5% (n = 18/68) of women in study group A, and 43.8% (n = 35/80) of women in study group B expressed their willingness to receive the vaccination (*p* = 0.029). There were no differences in willingness to receive the vaccination between different age strata of women in study groups A and B. (D) Conclusion: Our qualitative data demonstrate a benefit from oral counseling in addition to written briefing in order to increase the willingness to receive the vaccination among women in puerperium.

## 1. Introduction

The ongoing global coronavirus disease 2019 (COVID-19) pandemic, due to infection with the severe acute respiratory syndrome-coronavirus-2 (SARS-CoV-2), has caused significant morbidity and mortality worldwide [1,2]. Apart from the primary respiratory disease that SARS-CoV-2 initiates which has been postulated from the very beginning [3], accumulating evidence indicates that the virus induces a systemic disease via its primary binding site, the ACE-2 receptor (angiotensin converting enzyme 2) [4,5]. This receptor is next to various other locations [6] present in the placenta, decidua and general endothelial cells, which seems to be one of the predominant factors posing pregnant women at risk for severe coronavirus disease [6,7]. Pregnant women have an increased likelihood to be admitted to the ICU to require mechanical ventilation or ECMO (aRR 3.0, 2.9 and 2.4, respectively) when compared to non-pregnant women with COVID-19 [8]. Additionally, possible direct and several indirect negative effects for the fetus during infection in pregnancy, such as an increased risk for preterm delivery (OR 1.48) and stillbirth (OR 2.36), have been acknowledged [9,10]. Therefore, pregnant women are a potentially vulnerable group that needs to be targeted by vaccination programs for optimal protection for the mother-to-be and the unborn. 

With the first vaccines against SARS-CoV-2 being available since December 2020, a stepwise vaccination program was started in Austria, aiming for a high-vaccination coverage of the entire population [11]. The vaccines authorized by the European Medicines Agency (EMA) and available for the widespread vaccination campaign in Austria were primarily the two messenger-RNA (mRNA)-based vaccines by Pfizer-BioNTech (BNT162b2) and Moderna (mRNA-1273), as well as the vector-based vaccine by AstraZeneca (ChAdOx1, Chimpanzee Adenovirus vaccine vector Oxford 1). All of these vaccines were proven to be safe and effective in the general population [12,13,14,15,16]. However, special cohorts such as pregnant and breastfeeding women were, as commonly accepted in vaccine development, not included in the initial licensing studies, thus, precluding them from early access to the vaccination. Nevertheless, there were no pre-clinical safety concerns nor any adverse events in those individuals who were unwittingly pregnant or conceiving during the trials [17]. Consequently, vaccination programs of pregnant and breastfeeding women worldwide were initiated, although being off-label used, predominantly with mRNA-based vaccines; this again confirmed the safety and effectiveness of the vaccine during pregnancy in large real-world datasets [18,19,20].

Despite vaccination initiatives, many women are hesitant when they are offered the COVID-19 vaccination whilst in the peripartum period. For influenza vaccinations, maternal concerns regarding possible negative effects on the newborn while breastfeeding have been identified as one reason for vaccine hesitancy [21]. In Switzerland, an online study found that only 38.6% of the breastfeeding women would have been willing to receive a COVID-19 vaccination after the first pandemic wave in the case of availability [22]. In contrast, it has been described that a physician’s recommendation for the vaccination can have a positive impact on the decision of each individual patient [21,23,24]. For HPV, Rosenthal et al. [23] reported that women who had a physician’s recommendation for the HPV vaccine, and who had the opportunity to talk about the vaccination with their physician personally, had a higher likelihood of receiving the vaccination.

Taken together, the goal of our study was to evaluate (I) the overall willingness of women to receive a COVID-19 vaccination in the early postpartum period, as well as to (II) evaluate whether a recommendation for the COVID-19 vaccination by a physician had an impact on this decision in postpartum women.

## 2. Materials and Methods

Ethics statement: This study was performed after the approval of the local ethics committee of the Medical University of Vienna (registration number 1208/2022). The study was conducted according to the Declaration of Helsinki, and all enrolled participants gave their oral and written consent to participate in the study. 

Study design: This was a qualitative two-stage prospective cohort study that was conducted at the Department of Obstetrics and Gynecology, Division of Obstetrics and Feto-Maternal Medicine, Medical University of Vienna (Vienna, Austria), between 1 October and 23 December 2021. Our hospital is a tertiary referral center for intermediate and high-risk pregnancies, which serves about 2800 deliveries per year, with referrals from all over the country. 

For our study, the women admitted to the two available postnatal wards during puerperium at our center were initially included in the evaluation group in order to assess the overall vaccination status and their willingness to receive the vaccination (time period: first three weeks of October). Subsequently, another cohort of unvaccinated women were randomly assigned to study groups A and B (time period: from the end of October until December). Study group A received a written briefing recommending the SARS-CoV-2 vaccination during puerperium. Women in study group B received the same written briefing and they had a 5 min oral counseling recommending the SARS-CoV-2 vaccination, conducted by an attending physician on the postpartum ward. The women were randomly assigned either to study group A or B, depending on the postpartum ward they were admitted to. This was carried out in order to prevent bias due to background risks or other factors in those women, as the admission to the two postnatal wards is random and the women do not differ in risk profile or other aspects of obstetric care. Women who were assigned to the evaluation group (unvaccinated, partially vaccinated and vaccinated women) received a written questionnaire assessing their vaccination status and their willingness to receive the vaccination. Women in study groups A and B were asked about their willingness to receive a vaccination during their postpartum stay, using the messenger-RNA (mRNA)-based vaccine by Pfizer-BioNTech (BNT162b2).

Evaluation group: The included women were either vaccinated, partially vaccinated (1 dose received) or unvaccinated. Women of this group received a written questionnaire, assessing their vaccination status and willingness to receive the vaccination. Women of the evaluation group were neither informed about the currently available recommendations for COVID-19 vaccines, nor were they personally counseled by a physician.

Study group A (written briefing only): Women of this group were handed an information sheet about the currently available recommendation for the vaccination against SARS-CoV2 during the postpartum period, following the guidelines of the Austrian Society for Gynecology and Obstetrics [25]. Women of study group A were exclusively given the written information about this recommendation without personal counseling by a physician. After receiving the information sheet, they were asked about their willingness to receive the SARS-CoV-2 vaccination during their postpartum stay via a questionnaire and were offered an opportunity to receive the COVID-19 vaccine onsite.

Study group B (written and oral briefing): Women of study group B were also handed the same information sheet, but they were additionally counseled by a physician about the purpose, potential risks and benefits of the COVID-19 vaccine. During this 5 min counseling, the women had the chance to ask questions about the vaccine, its potential maternal and neonatal effects. This was followed by a questionnaire assessment about their willingness to receive the vaccination, and were offered the opportunity to receive the COVID-19 vaccine onsite. 

Statistical analysis: Descriptive data were described as the median and interquartile range (IQR) for continuous variables. Categorical variables were specified as numbers (n) and percentages (%). In order to compare groups with categorical variables, we used the Pearson’s chi-squared test. For evaluating imbalances in the baseline characteristics between women in the three different cohorts, a standardized difference was calculated by subtracting the mean value of one cohort from the mean value of another cohort, and dividing the result by the standard deviation of the overall cohort. In accordance with Cohen et al. [26], an effect size index of greater than 0.2 was used to determine differences between cohorts (small-effect size). A two-sided *p*-value of *p* < 0.05 was considered statistically significant. For the analyses, we used IBM SPSS Statistics 27 (IBM, Armonk, New York, USA), and for the graphs, we used GraphPad Prism 8 (Graphpad Software, La Jolla, CA, USA).

## 3. Results

### 3.1. Patient Characteristics (Table 1, Figure 1)

We screened a total of 217 postpartum women with a median age of 31.5 years. Out of these, 69 women (31.8%) were included in the evaluation group, and 148 women (68.2%) were enrolled for the study, whereof 68 women (45.9%) had a written briefing only (study group A), and 80 women (54.1%) had an additional oral briefing (study group B). The patient flowchart of the evaluation group and the study groups A and B is outlined in Figure 1.Figure 1Patient flowchart. A total of 217 women in puerperium were included. A total of 69 (31.8%) patients were assigned to the evaluation group and received written questionnaires. A total of 148 patients were assigned to study groups A and B. Patients in study group A (n = 68, 31.3%) were handed out a written briefing recommending the SARS-CoV-2 vaccination during puerperium, while 80 (36.9%) patients were assigned to study group B and received an oral recommendation to undergo the SARS-CoV-2 vaccination during puerperium in addition to the written briefing.
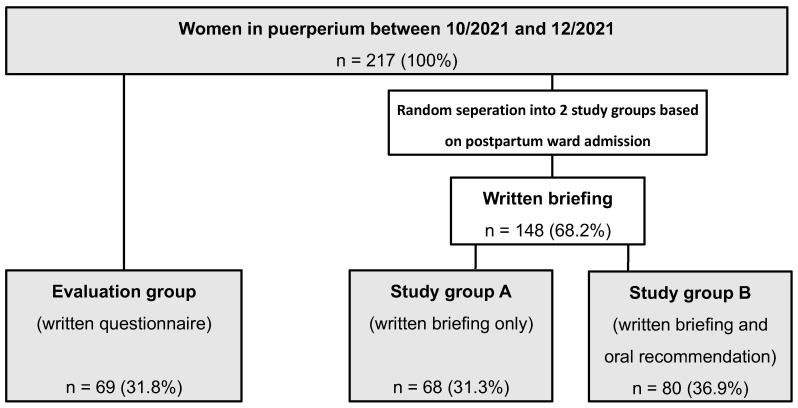


The median age (evaluation group: 31.5 years vs. A: 32.6 years vs. B: 32.5 years), migration background (evaluation group: 53.6% vs. A: 58.8% vs. B: 61.3%; *p* = 0.635) and delivery mode (vaginal—evaluation group: 52.2% vs. A: 50.7% vs. B: 51.2%; cesarean—evaluation group: 44.9% vs. A: 46.3% vs. B: 45.0%, instrumental—evaluation group: 2.9% vs. A: 3.0% vs. B: 3.8%) were similar between these groups. The prevalence of preterm delivery was 26.1% in the evaluation group, 13.2% in study group A and 18.8% in study group B. This represents an imbalance, particularly between the evaluation group and study group A (standardized difference −0.32). Moreover, there were more women who were primipara in the evaluation group, as compared to study groups A and B (A: 44.1% vs. B: 37.5% vs. evaluation group: 63.8%; standardized difference—A vs. evaluation group: −0.39, B vs. evaluation group: −0.49). The sociodemographic maternal characteristics are shown on Table 1.
vaccines-10-01505-t001_Table 1Table 1Sociodemographic characteristics of 217 women enrolled in the evaluation group and the study groups A and B.
Evaluation Group(n = 69)Study Group A(n = 68)Study Group B(n = 80)Standardized Difference ^1^Evaluation Group vs. AEvaluation Group vs. BA vs. BAge, years (IQR)31.5 (28.1–36.4)32.6 (28.9–36.0)32.5 (28.8–35.7)−0.03−0.02−0.01Migration background, n (%)37 (53.6%)40 (58.8%)49 (61.3%)0.110.15−0.05Primipara, n (%)44 (63.8%)30 (44.1%)30 (37.5%)−0.39−0.490.11Mode of delivery


0.030.03−0.00Vaginal delivery, n (%)36 (52.2%)34 (50.7%)41 (51.2%)


Cesarean section, n (%)31 (44.9%)31 (46.3%)36 (45.0%)


Instrumental delivery, n (%)2 (2.9%)2 (3.0%)3 (3.8%)


Preterm delivery, n (%)18 (26.1%)9 (13.2%)15 (18.8%)−0.32−0,19−0.14^1^ Imbalance defined as absolute value greater than 0.20 (small-effect size).

### 3.2. Women’s Assessment on Their SARS-CoV-2 Vaccination Status (Table 2, Figure 2)

Out of the 69 women in the evaluation group, 23 women (33.3%) received one shot of the COVID-19 vaccine, and 22 women (31.9%) were fully vaccinated by receiving two shots. Among unvaccinated women in the evaluation group, 18/46 women (39.1%) stated that they would prefer to receive the COVID-19 vaccine in puerperium during their hospital stay if given the possibility. Among unvaccinated women who declined the SARS-CoV-2 vaccination in puerperium, 25.0% (n = 7/28) of women reported that they planned to receive the vaccination in the future after hospital dismissal, while 75.0% (n = 21/28) categorically declined to receive the SARS-CoV-2 vaccination both in puerperium and in the future. Finally, of 19 women who had their first vaccination in puerperium, 14 (73.7%) reported that they would receive a second dose of the SARS-CoV-2 vaccination after hospital dismissal. Table 2 and Figure 2 summarize the vaccination status and willingness to receive the vaccination among women in the evaluation group.vaccines-10-01505-t002_Table 2Table 2Assessment of SARS-CoV-2 vaccination status and willingness to receive the vaccination among 69 women in the evaluation group.
YesNoReceived at least one shot of the SARS-CoV-2 vaccination23 (33.3%)46 (66.7%)Received two shots of the SARS-CoV-2 vaccination22 (31.9%)47 (68.1%)Unvaccinated and want to receive the SARS-CoV-2 vaccination during puerperium/total number of unvaccinated women18/46 (39.1%)28/46 (60.9%)Did not receive the SARS-CoV-2 vaccination, but plan to receive the vaccination/total number of unvaccinated women declining the vaccination in puerperium7/28 (25.0%)21/28 (75.0%)One-time vaccinated * and want to receive a second shot after hospital dismissal/total number of one-time vaccinated women in puerperium14/19 (73.7%)5/19 (26.3%)* one-time vaccinated, either before or during puerperium.
Figure 2In the evaluation group (n = 69), of women in puerperium, their SARS-CoV-2 vaccination status and willingness to receive the vaccination was assessed via a written questionnaire. Overall, 33.3% were vaccinated with at least one shot, while 66.7% were not vaccinated. Of the n = 46 unvaccinated patients, 39.1% were willing to receive a vaccination during puerperium, whereas 60.9% stated they would decline the SARS-CoV-2 vaccination during puerperium. Of the 28 unvaccinated patients who declined the SARS-CoV-2 vaccination in puerperium, 25% (n = 7) had plans on receiving the SARS-CoV-2 vaccination later on, while 75.0% (n = 21) declined the SARS-CoV-2 vaccination in general.
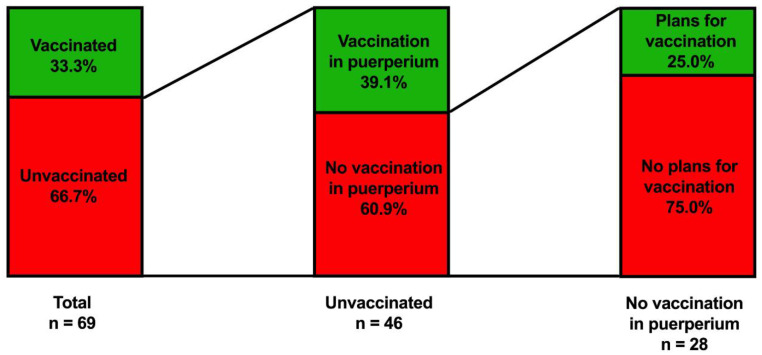


### 3.3. Effect of Physician’s Counseling on SARS-CoV-2 Vaccination (Figure 3)

Out of the 68 women in study group A, 18 women (26.5%) asked for a vaccine postpartum, whereas 35 of the 80 women (43.8%) of study group B asked for the vaccination (*p* = 0.029). This significant difference in willingness to receive the vaccination is also depicted in Figure 3.Figure 3Comparison of the rate of willingness to receive the SARS-CoV-2 vaccination among women in study group A (written briefing only) and study group B (written briefing and oral recommendation) during puerperium. Women in study group B were more likely to receive the vaccination after giving birth than women in study group A (*p* = 0.029).
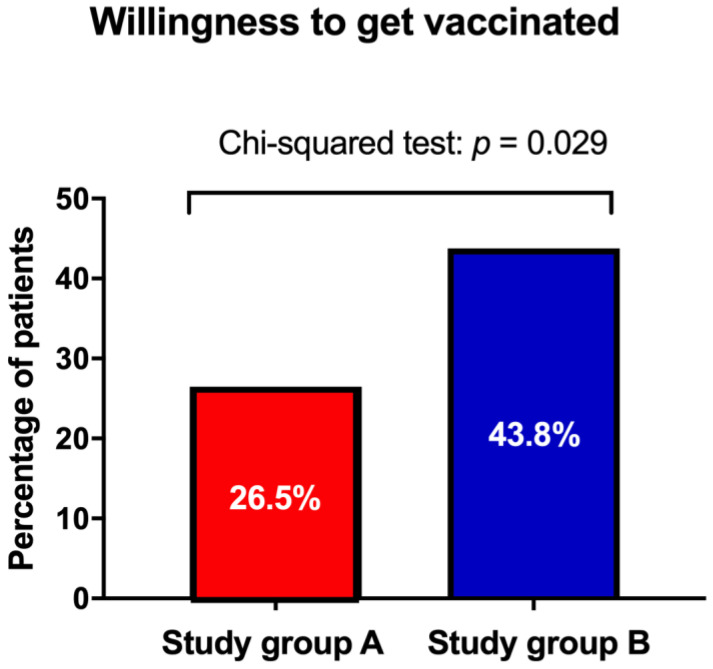


### 3.4. Willingness to Receive the Vaccination in Different Age Strata of Women in Puerperium (Table 3, Figure 4)

Table 3 illustrates the number of women who were unvaccinated in the evaluation group and women who had either a written briefing (study group A) or written and oral briefing (study group B) regarding their wish to receive the vaccination divided into four different age groups (A: 18–29 years; B: 30–39 years; evaluation group: 40–49 years).
vaccines-10-01505-t003_Table 3Table 3Willingness to receive the vaccination among different age groups of study group A, study group B and unvaccinated women of the evaluation group.Age Groups(years)Overall(n = 194)Study Group A(n = 68)Study Group B(n = 80)Unvaccinated Women in the Evaluation Group(n = 46)YesNoYesNoYesNoYesNo**18–29**29(40.8%)42(59.2%)4(17.4%)19(82.6%)12(44.4%)15(55.6%)13(61.9%)8(38.1%)**30–39**43(39.8%)65(60.2%)14(34.1%)27(65.9%)20(43.5%)26(56.5%)9(42.9%)12(57.1%)**40–49**6(40.0%)9(60.0%)0(0%)4(100%)3(42.9%)4(57.1%)3(75.0%)1(25.0%)

Considering all unvaccinated women included in this study, the willingness to receive the vaccination was almost identical among different age groups (18–29 years: n = 29/71 (40.8%) vs. 30–39 years: n = 43/108 (39.8%) vs. 40–49 years: n = 6/15 (40.0%)).

Women of study group A, who were younger than 40 years old, seemed to be more likely to receive the vaccination (18–29 years: n = 4/23 (17.4%) vs. 30–39 years: n = 14/41 (34.1%) vs. 40–49 years: n = 0/4 (0%)). Conversely, there was no difference in willingness to receive the vaccination between different age strata in study group B (18–29 years: n = 12/27 (44.4%) vs. 30–39 years: n = 20/46 (43.5%) vs. 40–49 years: n = 3/7 (42.9%), see Figure 4).Figure 4Stratification of the patients in different age groups of (**A**) the overall study group collective (i.e., study group A and B), (**B**) study group A and (**C**) study group B. There was no difference between age groups in the overall study group collective or study group B. In study group A (written briefing only), the group of women aged 40–49 showed no interest in receiving or planning to receive the SARS-CoV-2 vaccination.



## 4. Discussion

In this prospective study, we were able to demonstrate that there are strategies to increase the willingness to receive the SARS-CoV-2 vaccination in women postpartum. By assessing the vaccination status of women at our postpartum ward, we observed a 33.3% rate of women who were fully vaccinated against SARS-CoV-2 as baseline. When women were counseled by a physician who explained the main risks and benefits of the COVID-19 vaccine, women were able to address their concerns and questions, which they found of high relevance to them. We believe that this strategy could potentially increase the perinatal COVID-19 vaccination rate.

Vaccination rates in Austria are among the lowest in Western Europe, with 73.2% of the general population being vaccinated in February 2022 [27]. Regarding pregnant and breastfeeding women, the Austrian Society for Obstetrics and Gynecology recommended the COVID-19 vaccination in March 2021, which stands in line with other international societies, although being off-label used during pregnancy. Shortly thereafter, the National Health Committee [11] prioritized pregnant women for COVID-19 vaccinations in the National Vaccination Program as one of the vulnerable groups that may face the negative effects of COVID-19 [11,25]. However, as illustrated by our results, the number of women who are willing to receive the vaccination during pregnancy and postpartum is rather low; out of our study cohort, only 33.3% of the women announced their interest to receive the vaccine. After they were handed an information sheet about the importance of the COVID-19 vaccination, 26.5% were willing to receive the vaccination. This number could be increased up to 43.8% after personal counseling by a physician.

This finding stands in accordance with previously published studies on pregnant and postpartum women, reporting that the SARS-CoV-2 vaccine acceptance rates were between 33% and 37% [28,29]. Reasons for this hesitancy are very heterogeneous. A lack of information about the vaccine’s safety during this vulnerable phase of life, concerns about the effectiveness of the vaccine itself, as well as concerns about neonatal health are considered to play a role in this context [28].

Women of a reproductive age who are planning to conceive need to be considered in times of a pandemic, when new treatment options or preventive measures, such as vaccines, are being developed [30,31]. It is of paramount importance to offer these women targeted and professional information [30], as there is otherwise a systematic inequality for this population [32]. As it has recently been highlighted by a USA survey [33], concerns about possible negative effects on the unborn, together with the scarce data, are among the predominant reasons for maternal vaccine hesitancy.

Our results are generally in line with those of previously published studies, reporting that personal contact and recommendation by a physician are crucial for the patient’s decision to receive the vaccination or not [23,34]. Moreover, it has been reported that the willingness to receive the vaccination against SARS-CoV-2 is higher in older individuals [35,36]. Among the postpartum women in our study, age did not correlate with vaccine acceptance and there was no significant difference with regard to the vaccination status between women of different age groups [29].

In an observational study, SARS-CoV-2 mRNA vaccines showed a stable maternal immune response with transplacental transfer of antibodies, detectable in cord blood after the first vaccination dose, indicating possible protection of the infant as well [37]. Moreover, SARS-CoV-2-specific immunoglobulin (Ig) A and Ig G antibodies are secreted in breast milk after the vaccination of SARS-CoV-2 mRNA for 6 weeks [38]. Antibodies in breast milk present strong neutralizing effects as well. Thus, evidence is accumulating that the vaccination induces a stable immune response in breastfeeding women with the secretion of antibodies into the breast milk [39,40,41]. No evidence is available pointing towards any harm from the vaccination for lactating women or their breastfed newborns. Vaccinations of pregnant (and breastfeeding) women are already a valuable tool for protecting the newborn in the first few months of life from severe adverse disease effects of other diseases, such as influenza [42,43]. Thus, the early breastfeeding period seems to offer an ideal time-point for vaccinating those who did not receive the vaccination during pregnancy, additionally as it is likely to offer the newborn a certain protection against infection in the first few months, especially concerning the fact that acceptance of the SARS-CoV-2 vaccination is low during pregnancy [29]. Therefore, the vaccination against COVID-19 in pregnant and breastfeeding women is strongly recommended by various international scientific societies of obstetrics and gynecology [25,44,45,46].

Although prospective, our study design had some limitations. Firstly, this was a single-center study, and sampling bias cannot be completely ruled out so that external validation of our findings is required. This bias, however, seems to be unlikely, as our findings are consistent with those of the existing literature in terms of which measures could help to improve vaccine acceptance in vulnerable patient groups [28,29]. Secondly, we did not assess the influence of language barrier and/or different cultural backgrounds, which might have an impact on the women’s decision [47,48]. Thirdly, we must admit that our findings need to be reevaluated in light of recent developments, such as novel SARS-CoV-2 variants and possibly new vaccinations. There was a difference between the evaluation group and the study cohorts regarding the prevalence of primiparity and preterm delivery, which represents a limitation. However, importantly, there were no differences in baseline characteristics between study group A and B. Moreover, oversampling was not considered for the evaluation group, which had a lower proportion of unvaccinated women (66.7%) as compared to study group A and B (100%). As a result, study groups A and B were different from *the evaluation group*, which is why these groups were not directly compared. Furthermore, no sample size or power analysis were performed for this study. Rather, all women in puerperium during the study period (October–December 2021) were evaluated for study inclusion. Women were randomly assigned to the study groups A and B depending on the postpartum ward they were admitted to. This represents a potential limitation of our study. However, at our center, physicians attending women in the postpartum wards are randomly assigned to the different wards for daily ward rounds and rotate on a day-to-day basis. Thus, the same physicians attend the two different postpartum wards on any given day. Further, the admission to one of the two wards happens randomly, thus, women presenting with different obstetric issues should be distributed rather equally across the wards (i.e., study groups), representing the obstetric population taken care of at our center. Finally, no data were collected after hospital dismissal, which represents another limitation.

## 5. Conclusions

In conclusion, this qualitative study found the ability of a personal 5 min counseling by a physician to increase the willingness to receive the vaccination against SARS-CoV-2. While the vaccination in pregnant and postpartum women is generally recommended, its acceptance is rather low in this cohort. The implementation of this measure could therewith increase the vaccination acceptance in this cohort, which stands in line with Nooney et al. [32], stating that “the health of the child begins with the health of the mother”.

This section is not mandatory but can be added to the manuscript if the discussion is unusually long or complex.

## Data Availability

The data presented in this study are available to all authors and can be provided upon reasonable request to the corresponding author.

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
