# Peer review of "Written Briefing and Oral Counseling Increase the Willingness to Receive the SARS-CoV-2 Vaccination among Women in Puerperium: A Qualitative Prospective Cohort Study"

_vaccines, 2022, doi:10.3390/vaccines10091505_

Round 1

Reviewer 1 Report

This article is a good OBGYN study for promoting vaccination among women in puerperium. However, the result part could be better with more data based details.

It would more interesting to show the Assessment of SARS-CoV-2 vaccination status and desire to get vaccinated of all groups in table 2. Did individuals in group A and B finish their questionaire after information delivery and counseling? Also the table 3 would be better, if the author could add the group C and overall willingness of all age groups of each group.

Reviewer 2 Report

The present manuscript presents results of a counseling intervention to increase willingness to receive a COVID-19 vaccine among women in puerperium. The study is important because it informs that interventions are or not effective and can be implemented with low cost in similar settings. There are several concerns on how the methods and results are presented.

If the study was design with randomization, even when this was done as a quality improvement exercise, please include this description in the abstract.

It is unclear whether women in interventions A and B (active intervention groups) had different criteria for selection, and thus could not be randomly assigned. Women in the control group (group C) had completed the primary vaccination scheme, received one dose, or unvaccinated, while women in groups A and B were unvaccinated prior to the hospitalization, or during the hospitalization. This needs to be clear to the reader.

It no randomization was performed to assign to group C, this needs to be described. It is unclear whether the study considered oversampling this group to overcome the percent that had already been vaccinated (33%) or getting the vaccine during the hospitalization (38%).

It is unclear whether there is planning to collect information of vaccination status after hospital dismissal. If this information could be retrieved, this would improve the quality of the paper.  If this is not possible, include this as a limitation.

If a sample size and power analysis was performed before the study was implemented, provide this information. If not, include this in the limitation section.

To characterize differences across intervention arms, avoid using p-values to define whether characteristics are similar. P-values depend on sample size (not powered), and prevalence. For example, a preterm delivery of 13% to 26% is clinically different. Use standardized difference to refer to meaningful differences when describing characteristics.

Results of Control group is confusing. It was unclear whether 21 reported that would accept a vaccine later, or it was 7 who reported that they would accept a vaccine later. Do not report differences by age strata. The sample size limits this analysis. Include this as a limitation.

Round 2

Reviewer 2 Report

The manuscript revised provided further details and corrections to the presentation of results. However, all changes seemed like a patch, without a rational of why to keep or exclude information in this new version.

Some examples

In the abstract, it was included that this is randomized, but my understanding of what they did was a two-stage study.  First, they evaluated the vaccination status among all eligible women, and second, among those unvaccinated, they randomized into three arms, A (written briefing), B(written and oral briefing), C (none). If not, please described how unvaccinated women were assigned to C

It is unclear why they collapsed vaccinated women with those randomized to C. This might be driving the differences in baseline characteristics in C as compared to the other two groups. I.e., vaccinated women (example, primipara).

If my understanding is correct, they randomized and assigned to A, 68 women, B, 80 women, and C , 46 women. What was the randomization method used that provided such an unbalanced sample size, or was not a randomization scheme used. Indicate the methods.

The authors have added that the study assigned women to arm C, to evaluate vaccination status and willingness to get vaccinated. If data are available for the first stage of the study (ie. Without the intervention), use these data to describe the flow on the distribution of vaccinated to plans for vaccination. The way is set up is biased, using only arm C reduces the number of unvaccinated artificially.

Statistical analysis now uses the suggestion to address imbalance using standardized difference. However, statistical tests are still described, but are not reported in the result section.

It is unclear why the overall willing to be vaccinated is not reported comparing to C, overall but it is reported by age group (as a table and a figure), where the sample is so limited. It is very difficult to make any consideration on the age strata.

Author Response

"Please see the attachment". 
